# Improving the Maternity Care Safety Net: Establishing Maternal Mortality Surveillance for Non-Obstetric Providers and Institutions

**DOI:** 10.3390/ijerph21010037

**Published:** 2023-12-27

**Authors:** Joan L. Combellick, Bridget Basile Ibrahim, Aryan Esmaeili, Ciaran S. Phibbs, Amanda M. Johnson, Elizabeth Winston Patton, Laura Manzo, Sally G. Haskell

**Affiliations:** 1Department of Veterans Affairs, Veterans Health Administration, Office of Women’s Health, 810 Vermont Ave NW, Washington, DC 20420, USA; amanda.johnson@va.gov (A.M.J.); elizabeth.patton@va.gov (E.W.P.); sally.haskell@va.gov (S.G.H.); 2VA Connecticut Healthcare System, 950 Campbell Ave, West Haven, CT 06516, USA; 3School of Nursing, Yale University, 400 West Campus Drive, Orange, CT 06477, USA; bridget.basileibrahim@yale.edu (B.B.I.);laura.manzo@yale.edu (L.M.); 4Health Economics Resource Center (HERC), Palo Alto VA Medical Center, Menlo Park 795 Willow Road, Palo Alto, CA 94025, USA; aryan.esmaeili@va.gov (A.E.); cphibbs@stanford.edu (C.S.P.); 5Departments of Pediatrics and Health Policy, Stanford University School of Medicine, 453 Quarry Road, Palo Alto, CA 94304, USA; 6VA Boston Health Care System, 150 South Huntington Avenue, Boston, MA 02130, USA; 7Department of Obstetrics and Gynecology, Chobanian & Avedisian School of Medicine, Boston University, 771 Albany St, Dowling 4, Boston, MA 02118, USA; 8US Army, AMEDD Student Detachment, 187th Medical Battalion, Joint Base San Antonio, San Antonio, TX 78234, USA; 9School of Medicine, Yale University, 333 Cedar St, New Haven, CT 06510, USA

**Keywords:** maternal mortality, pregnancy-associated mortality, maternal outcomes, veterans, pregnancy outcomes, pregnancy, high risk, epidemiologic surveillance

## Abstract

The siloed nature of maternity care has been noted as a system-level factor negatively impacting maternal outcomes. Veterans Health Administration (VA) provides multi-specialty healthcare before, during, and after pregnancy but purchases obstetric care from community providers. VA providers may be unaware of perinatal complications, while community-based maternity care providers may be unaware of upstream factors affecting the pregnancy. To optimize maternal outcomes, the VA has initiated a system-level surveillance and review process designed to improve non-obstetric care for veterans experiencing a pregnancy. This quality improvement project aimed to describe the VA-based maternal mortality review process and to report maternal mortality (pregnancy-related death up to 42 days postpartum) and pregnancy-associated mortality (death from any cause up to 1 year postpartum) among veterans who use VA maternity care benefits. Pregnancies and pregnancy-associated deaths between fiscal year (FY) 2011–2020 were identified from national VA databases. All deaths underwent individual chart review and abstraction that focused on multi-specialty care received at the VA in the year prior to pregnancy until the time of death. Thirty-two pregnancy-associated deaths were confirmed among 39,720 pregnancies (PAMR = 80.6 per 100,000 live births). Fifty percent of deaths occurred among individuals who had experienced adverse social determinants of health. Mental health conditions affected 81%. Half (n = 16, 50%) of all deaths occurred in the late postpartum period (43–365 days postpartum) after maternity care had ended. More than half of these late postpartum deaths (n = 9, 56.2%) were related to suicide, homicide, or overdose. Integration of care delivered during the perinatal period (pregnancy through postpartum) from primary, mental health, emergency, and specialty care providers may be enhanced through a system-based approach to pregnancy-associated death surveillance and review. This quality improvement project has implications for all healthcare settings where coordination between obstetric and non-obstetric providers is needed.

## 1. Introduction

It is currently more dangerous to give birth in the United States (U.S.) than in any other high-income country, with four out of five maternal deaths deemed preventable [1]. Though many public health initiatives have been launched to address this public health crisis, the maternal mortality ratio has continued to increase. In 2021, the maternal mortality ratio was 32.9 per 100,000, with significant racial disparities in outcomes [2]. According to the U.S. Centers for Disease Control and Prevention (CDC), the maternal mortality rate for non-Hispanic Black individuals was 2.6 times higher than the rate for non-Hispanic White women in 2021 [2].

A life course approach has been prioritized to address the maternal mortality crisis in the U.S. [3]. This approach acknowledges the impact of social and environmental contextual factors that are outside the healthcare system and less proximal to the pregnancy that negatively impacts maternal and infant outcomes. Nonetheless, recommendations arising from maternal mortality review committees often focus on the role of obstetric providers, with less attention paid to developing capacity among primary, mental health, emergency, and specialty care providers. Optimizing and broadening services—from pre-conception through pregnancy and postpartum—among all healthcare providers and institutions is a key strategy for improving maternal outcomes.

Additionally, developing a granular view of the unique risk factors that affect specific high-risk populations has been identified as an important strategy for improving maternal outcomes [4]. Not only does this reveal disparities and structural inequities related to adverse outcomes, but it also provides a more actionable approach to improving care for that population. Care improvements can be initiated at the provider, institution, or health system level, and all these levels are relevant to mortality surveillance.

In response to these recommendations, the VA launched a maternal mortality surveillance committee to review all pregnancy-associated deaths among veterans who received VA maternity care benefits to cover their costs with community providers. These individuals received their maternity care from diverse providers and institutions across the United States (US) but continued to access primary, mental health, emergency, and specialty care at the VA.

The population of reproductive-aged veterans is growing rapidly, with a high incidence of risk factors known to be associated with adverse maternal and neonatal outcomes. Among eligible and enrolled veterans, use of VA maternity care benefits increased more than 14-fold between 2010 and 2015, with use by those 35 years or older increasing 16-fold during that period [5]. In 2020, the VA was providing coverage for approximately 4000 pregnancies per year [6]. When compared to the general population, veterans using VA benefits are more likely to experience hypertensive disorders of pregnancy, gestational diabetes [7], and a body mass index (BMI) classified as overweight (BMI = 25–<30 kg/m^2^) or obese (BMI ≥ 30 kg/m^2^) [8,9]. Pregnant veterans are also twice as likely as their non-pregnant counterparts to have a mental health diagnosis [10], and around 30% of those using maternity benefits report needing mental health services during pregnancy [6]. Rates of post-traumatic stress disorder (PTSD) are higher among veterans as compared to civilians [11], and PTSD has been associated with increased preterm birth [12], gestational diabetes, preeclampsia [13], postpartum depression/anxiety, and self-perception of experiencing a difficult pregnancy [14].

Some groups at increased risk of adverse outcomes are overrepresented in the Veteran population, including Black or African American (hereafter referred to as Black) and American Indian/Alaska Native individuals; those who reside in rural areas; and those who are 35 years old or older [5]. System-level factors that may affect veterans include issues related to care coordination between VA and community facilities [15], lack of training in veteran-focused care on the part of community providers [16,17], and other factors such as structural racism implicated in disparities nationally [18].

Current pregnancy support at the VA includes a national corps of Maternity Care Coordinators, composed of nurses, social workers, and others who provide support, outreach, and care coordination to veterans via in-person visits, telephone calls, and/or video connection during and after pregnancy [19]. This program was initiated in 2012 and was developed to support veterans who are receiving care from dual healthcare systems. The program has grown to extend coverage nationally and, in 2023, was extended to provide support through the first year post-partum rather than the first 42 days only. Maternity Care Coordinators help patients access care and resources both at VA and with their community provider. They also provide individual continuity and emotional support, factors that have been associated with improved maternal outcomes. The maternal mortality review committee sought to build on the services provided by Maternity Care Coordinators to improve surveillance capacity, identify individual and population-level risk factors of pregnant veterans, improve care coordination, and provide recommendations for primary and mental health care providers.

This project aimed to (1) report maternal mortality and pregnancy-associated mortality among veterans utilizing VA maternity benefits (fiscal year 2011–2020); (2) describe the process of establishing a maternal mortality review committee within a healthcare system that does not directly provide maternity care; and (3) present relevant recommendations to improve non-obstetric healthcare services before, during, and after pregnancy. This quality improvement project was undertaken and approved by the VA Office of Women’s Health. IRB approval was not required. (Please see Appendix A).

## 2. Materials and Methods

This retrospective analysis included all pregnancies and pregnancy-associated deaths among veterans using VA maternity care benefits from fiscal year 2011–2020. Each mortality case underwent individual medical record review.

**Identification of mortality cases.** Inclusion criteria for this evaluation of pregnancy-associated mortality among veterans between FY 2011–2020 included:Eligibility for benefits in the Department of Veterans Affairs (VA);Enrollment in VA;The use of VA maternity care benefits (as opposed to a third-party payer, such as Medicare or private insurance through an employer or spouse).

Exclusion criteria included:Duplicate records;Lack of active engagement in VA care, operationalized as pregnancies with no evidence of a VA primary care visit in the year prior to the veteran’s calculated last menstrual period.

Encounters from all veterans during the specified time period with evidence of pregnancy outcomes were extracted from VA billing data using international classification of disease (ICD)-9/10, diagnosis-related group (DRG), and current procedural terminology (CPT) codes. This included any pregnancy outcome (pregnancy loss and live births at any gestational age). Billing data, derived from claims submitted from non-VA providers and hospitals for services rendered to veterans, were also used to identify pregnancy outcomes. After limiting by inclusion and exclusion criteria, deaths were identified using the VA Vital Status Mast File (VSF), which includes national data from the Veterans Benefits Administration Integrated Benefits System (IBS) Death File, Medical Inpatient Datasets, the Social Security Administration (SSA) Death File, and the Medicare Vital Status File. A second VA-based research group was approached to validate the algorithm for identifying pregnancy-associated deaths developed by the first group. The results of this assessment were largely concordant. The mortality identification algorithm was subsequently revised to include search strategies from both groups.

*Pregnancy-associated death* was defined as death “during pregnancy or within 1 year of the end of a pregnancy from any cause” [20]. This differs from the definition of *pregnancy-related death*, which is defined as death “during pregnancy or within one year of the end of pregnancy from a pregnancy complication, a chain of events initiated by pregnancy, or the aggravation of an unrelated condition by the physiologic effects of pregnancy” [21]. *Maternal mortality* was defined as death “while pregnant or within 42 days of termination of pregnancy, irrespective of the duration and the site of the pregnancy, from any cause related to or aggravated by the pregnancy or its management, but not from accidental or incidental causes” [22].

**Mortality Review.** All deaths identified between FY 2011–2020 received an in-depth review by a clinician-researcher trained to ascertain the cause and timing of death. To calculate the maternal mortality ratio (which specifies death must be pregnancy-related), deaths identified during pregnancy or within the first 42 days postpartum underwent a confirmatory review by a second trained clinician to determine if the death was related to pregnancy or from an incidental or accidental cause. If the cause of death could not be identified solely from the medical record, further information was sought through the National Death Index or via local medical examiners’ offices.

The individual review was completed using a chart abstraction template that was based on the CDC mortality review guidelines with additional veteran-specific factors. Charts were accessed through Joint Longitudinal Viewer (JLV), VA’s VistAWeb chart review tool, which allows for the review of data generated from the VA, the Department of Defense, and community care records.

We assessed demographics following standard VA reporting categories, including self-reported age, ethnicity (Hispanic/Latino, non-Hispanic/Latino), and race (Black/African American, White, Asian, Native Hawaiian and other Pacific Islander, and American Indian/Alaskan Native). Rurality was determined by using the VA designation of rural/very rural versus urban by zip code at the time of pregnancy and subsequently validated using the Rural Health Information Hub, a service designed to determine whether a specific location qualifies as rural based on eligibility for federal programs [23]. Gestational week of pregnancy at delivery, cause of death, timing of death in relation to pregnancy outcome, complications affecting pregnancy (including, but not limited to, hypertensive disorders of pregnancy, gestational diabetes, BMI ≥ 30), and mental health conditions (including, but not limited to, anxiety, depression, PTSD, and substance use disorder) were recorded.

Social determinants of health were assessed during individual chart review based on findings during the year prior to pregnancy, through pregnancy, until the time of death. Housing status was determined by (1) any mention of being unhoused or experiencing unstable housing in the medical record or (2) participation in the Housing and Urban Development VA Supportive Housing program (HUD-VASH), a collaborative program designed to meet the housing needs of veterans. Exposure to community violence or exposure to domestic violence was assessed via individual chart review and/or results of intimate partner violence screening in the medical record. Financial insecurity was assessed based on the mention of financial needs relating to transportation, nutrition, or childcare in the medical record.

**Multidisciplinary Review Committee**. A multidisciplinary maternal mortality review committee was developed to systematically review cases with a special focus on care provided by VA-based providers in the year prior to pregnancy, through pregnancy, and until the time of death, in addition to scanned records from maternity care providers in the community which are archived in the medical record. The committee systematically evaluated mortality cases to identify strengths and opportunities for care improvements in VA care. Each aspect of care provision (e.g., primary care, pharmacy services, and mental health care) was evaluated to determine if care deficits contributed to preventable mortality. Finally, a summative evaluation was done to identify prominent areas for improvement and successful practices that should be continued at the provider, institution, and healthcare system levels. The team included 10 individuals with expertise in obstetrics and gynecology, primary care, psychiatry, outreach and care coordination, social work, and maternal mortality. Reviewers were either exclusively based at the VA or dually appointed to academic institutions and the VA.

## 3. Results

**Maternal mortality and pregnancy-associated mortality**: Between VA Fiscal Year 2011–2020, a total of 39,720 pregnancies were evaluated; among them, 32 pregnancy-associated deaths were identified. The overall pregnancy-associated mortality ratio (PAMR) was 80.6 deaths per 100,000 live births among veterans using maternity care benefits paid for by the VA. Sixteen deaths occurred either during pregnancy or within the first 42 days postpartum. Fifteen of these were identified as cases that related to or were aggravated by the pregnancy but not from accidental or incidental causes. Thus, the maternal mortality ratio was 37.8 deaths per 100,000 live births.

**Demographics and access**: Of the 32 pregnancy-associated deaths, 10 (31%) were among veterans who were 35 years or older. The PAMR was highest among veterans in this age group at 96.5 deaths per 100,000 live births. The PAMR was higher for non-white veterans (116.1/100,000) than for White (78.5/100,000) veterans. The PAMR was higher among non-Hispanic veterans (85.2/100,000) than Hispanic veterans (62.6/100,000). Fewer veterans residing in rural areas died of pregnancy-associated events when compared to those living in urban areas (See Table 1).

**Mental health and chronic conditions:** Most veterans who died in pregnancy-associated events had at least one mental health condition during pregnancy (n = 26, 81%). Of these, 22 (85%) had two or more mental health comorbidities, and 21 (81%) were actively prescribed psychotropic medications during or after the index pregnancy. Approximately half of those who died, 15 (47%), had experienced traumatic events such as intimate partner, domestic, or community violence or had previously experienced physical or sexual abuse. Ten (31%) individuals had a diagnosis of substance or alcohol use disorder, and 12 (38%) smoked during pregnancy. A BMI of 30 or greater and hypertension affected 14 (44%) and 11 (34%) individuals, respectively. There were no individuals with a diagnosis of gestational diabetes. (See Figure 1). 

**Cause and timing of death**: The most frequent causes of death were cardiac arrest or cardiomyopathy (n = 5), overdose (n = 4), suicide (n = 4), homicide (n = 3), aneurysm (n = 3), and pneumonia (n = 3) (See Figure 2). Cause of death varied temporally, with suicide, overdose, and homicide more prevalent in the late postpartum period (43–365 days postpartum) and acute medical events occurring during pregnancy through postpartum day 42 (See Figure 2). Half (n = 16, 50%) of all deaths occurred in the late postpartum period. The remainder occurred during pregnancy or the first day following birth (n = 5, 15.6%) or the early postpartum period from 2–42 days postpartum (n = 11, 34.4%). (See Figure 2).

**Social Determinants of Health.** Social determinants of health with the potential to affect pregnancy outcome (being unhoused or with unstable housing, exposure to violence at home or in the community, and financial instability) affected half of those who died (n = 16, 50%) and were more likely to affect those who died in the late postpartum period between 43–365 days postpartum (n = 10, 63%).

**Care Coordination:** Of the 32 veterans who died in pregnancy-associated events, 13 (41%) had contact with a VA Maternity Care Coordinator. The frequency of involvement varied as some individuals had many points of contact throughout the pregnancy, while others had initial outreach only.

## 4. Discussion

The VA maternal mortality review committee, established within an organization that does not directly provide obstetric care, is an effective strategy for collecting data, providing ongoing surveillance, and developing recommendations for the non-obstetric care team, including primary, mental health, emergency, and specialty providers. It has provided insights into opportunities for care at the provider, institution, and system levels. Clinical implications identified by the maternal mortality review process are detailed below.

**Mortality data:** Maternal mortality (37.8 per 100,000) and pregnancy-associated death (80.6 per 100,000) were high among veterans using VA maternity benefits between FY 2011–2020. This is above the national maternal mortality ratio of 32.9 per 100,000 live births (2021) [2] and a semi-national (22-state) analysis that reported a pregnancy-associated mortality ratio of 42.3 per 100,000 live births [24]. However, it should be noted that the number of veterans giving birth each year is very small, and this limits the ability to meaningfully compare these outcomes to other populations, including the U.S. general population, where a little over 3.6 million births occur each year [25].

**Implications of demographics and access**: Veterans from minoritized racial groups, including Black/African American, American Indian/Alaska Native, Asian, and Native Hawaiian/Pacific Islander, had a significantly higher pregnancy-associated mortality ratio as compared to Whites in our cohort, even though the maternal mortality ratio was relatively similar between the two groups. Because numbers are small, it is difficult to draw definitive conclusions. However, the difference may reflect that minoritized individuals in this cohort were temporally more likely to die in the late postpartum period when adverse social determinants of health were also more likely to impact outcomes. This underscores the important role primary, mental health, and other non-obstetric care providers can play during the late postpartum period to address broader reproductive health issues and social determinants of health. At the institutional level, it reinforces the need for substantive support to alleviate adverse social determinants of health, such as housing and transportation, and ongoing efforts to address systemic racism, such as hiring culturally concordant providers and ensuring equal access to care.

**Implications of mental health conditions**: Mental health conditions were the most common pregnancy complication. While mental health conditions affect 52% of the general female veteran population [5], 81% of the veterans who died in pregnancy-associated events had been diagnosed with one or more mental health conditions. It is also striking that one-third (34%) of pregnancy-associated deaths among veterans were due to suicide, homicide, or overdose. This is higher than reported in a study of the U.S. general population (n = 11,782) in which drug-related deaths, homicide, and suicide accounted for 22.2% of pregnancy-associated deaths (2010–2020) [26]. These findings highlight the risk that mental health conditions pose during pregnancy and the postpartum period and underscore the need for targeted interventions to support individuals who are contending with mental health conditions. This also suggests the importance of developing mental health services that are easily accessed by reproductive-aged individuals. Finally, it was noted during chart review that mental health care providers often maintained continuity with high-risk veterans but usually did not provide pregnancy-related education or reproductive planning. Expanding the range of topics covered by mental healthcare providers to include plans for psychotropic medication management during and after pregnancy, contraception, childbirth preparation, lactation, and parenting could strengthen the maternity care support network for these individuals. Improving integration of mental health care with maternity care services through improved communication, collaboration, and, when feasible, co-location of services can also support seamless transitions for patients between specialties.

**Implications of chronic conditions:** Over a third of pregnancy-associated deaths were due to acute events related to cardiovascular conditions, aneurysms, and pulmonary embolus. Hypertensive disorders in pregnancy (including pre-pregnancy and pregnancy-associated hypertension) affected 34% of the cohort, while obesity (BMI ≥ 30) affected 44%. Hypertensive disorders of pregnancy are increasing nationally and differentially affect Black and Indigenous (American Indian/Alaska Native) individuals and those who are 35 years old or older [27]. Strategies to address and prevent maternal deaths from these acute events include early recognition and prompt treatment, along with increasing clinician and patient awareness of warning signs related to acute events [27]. Stabilizing chronic conditions and providing patient education are important preconception activities.

**Implications of timing of death and social determinants of health**: In our cohort, half (50%) of deaths from any cause and nearly all deaths by homicide, suicide, or overdose occurred in the late postpartum period (43–365 days postpartum) after maternity care in the community had ended. These findings reflect the vulnerability of the entire first year after birth as individuals adapt to new responsibilities and changing family structures. As mentioned, adverse social determinants of health, including housing instability, exposure to violence, and financial constraints, were also more likely to affect those who died in the late postpartum period. Primary care and mental health care providers who resume care after patients transition away from their maternity care providers may not be aware of pregnancy complications and may lack protocols that identify the full postpartum year as a time of elevated risk. It is crucial that healthcare providers understand the risks of the entire postpartum year as well as the social challenges faced by patients and implement increased surveillance and support, including referrals as needed between primary, specialty, and mental health care.

**Implications for Care Coordination**: In our cohort of pregnancy-associated deaths, care coordination was evident through the VA-based Maternity Care Coordinator program. This coordination often involved connecting veterans with supplies and pharmaceuticals, confirming referrals, making appointments, and resolving coverage problems. While Maternity Care Coordinators provide direct outreach to patients and facilitate referrals as needed, they do not orchestrate care at the provider level. There was little evidence in the medical record of consultation between providers in different specialties (such as mental health care and primary care) or between providers based at the VA and those based in the community. Developing improved communication between providers and institutions has been recognized as a key quality improvement measure to reduce pregnancy-associated mortality [21]. Segmented care has been identified as a contributor to maternal mortality in both high and low-income countries, and a framework for integration has been developed to minimize fragmentation of care and improve outcomes [28]. Barnea and colleagues provide examples of improved outcomes when transitioning from segmentation to integration of care from both small (Lithuania) and large (UK and South Africa) countries, aspects of which might be beneficial to adopt in the VA Care Coordination model [28].

Care coordination has been associated with greater patient satisfaction, decreased healthcare costs, and fewer medical errors [29]. However, further research is needed to assess the full impact of care coordination for veterans during pregnancy and postpartum, including on patient and provider satisfaction and subsequent uptake of services [29]. Maternity Care Coordinators are well-positioned to provide front-line data collection and rapid response to developing medical, emotional, and psychological problems. Care coordination at VA previously continued through the first 6 weeks postpartum only, but in response to heightened awareness of ongoing risk, this program was expanded in 2023 to allow coordinators to continue services through the first postpartum year.

## 5. Limitations

It is essential to consider the limitations of the data when interpreting these results. All case review information was derived from medical records archived within the VA electronic health record, including scanned documents from community care providers. These documents from community providers were requested for billing rather than clinical purposes and were submitted on a voluntary rather than mandatory basis. Thus, community care records were not consistently available. Our findings incorporated both ICD-9 and ICD-10 CM/PCS; thus, there may be limitations related to disruptions caused by this coding transition. Finally, because pregnancy-associated death is rare and this study is restricted to only veterans using VA maternity care benefits, our final sample size of 32 deaths is small. This limits any ability to draw definitive conclusions about associated risk factors and compare mortality ratios. It is also not possible to generalize these findings to other populations, including veterans who are not enrolled for care at the VA or those who use other coverage.

## 6. Conclusions

The findings of this VA-based maternal mortality review of non-obstetric care underscores the need for services tailored to veterans before, during, and after pregnancy. Improved integration of VA-based and community-based care would improve treatment continuity and access to veteran-specific care, especially in the area of mental health, which is by far the leading underlying pregnancy complication for individuals who die in pregnancy-associated events. The VA also has a significant opportunity for preconception care in the management of chronic conditions such as hypertension, obesity, mental health conditions, and substance use disorders. Patient education, preconception counseling, contraception management, and pregnancy options counseling are all important clinical foci for primary, mental health, emergency, and specialty care providers as the number of reproductive-age veterans continues to grow at VA. Ongoing surveillance by the multidisciplinary VA-based maternal mortality review committee will continue to provide policy and practice recommendations. Systematic surveillance and review of maternal deaths in institutions that do not provide maternity care is a productive means to develop and refine care interventions that strengthen the healthcare safety net and improve maternal outcomes.

## Figures and Tables

**Figure 1 ijerph-21-00037-f001:**
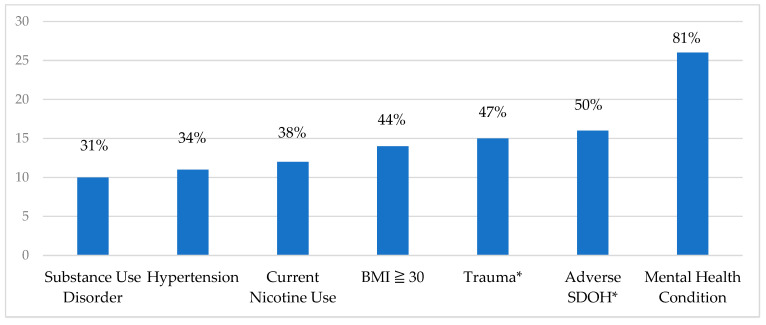
Conditions affecting pregnancy prior to death among veterans using VA maternity benefits, FY 2011–2020, (N = 32). Abbreviations: VA, US Department of Veterans Health Affairs; BMI, body mass index. Substance Use Disorder (includes Alcohol Use Disorder). * Trauma (includes exposure to interpersonal, domestic, and/or community violence). *Adverse Social Determinants of Health (SDOH) (includes housing, food, transportation, financial insecurity, or exposure to domestic or community violence).

**Figure 2 ijerph-21-00037-f002:**
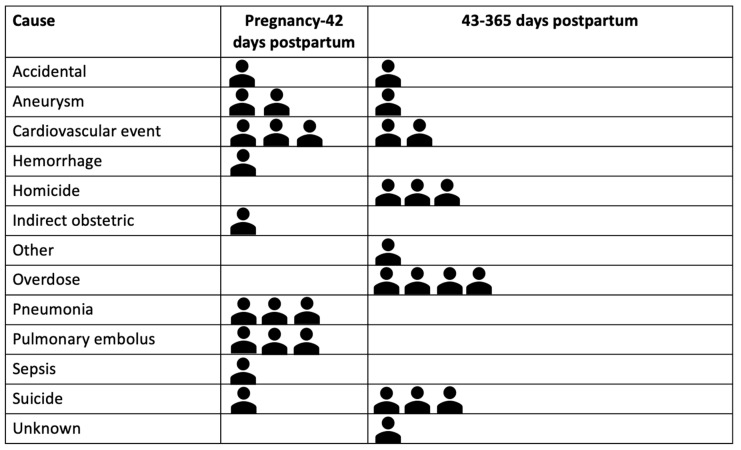
The cause and timing of pregnancy-associated deaths among veterans using VA maternity benefits, FY 2011–2020 (N = 32). Abbreviations: VA, US Department of Veterans Health Affairs.

**Table 1 ijerph-21-00037-t001:** Live births, maternal deaths, maternal mortality ratio, and pregnancy-associated mortality ratio among veterans using VA maternity care benefits, FY2011–2020, (N = 32).

Characteristic	Live Births (n)	Pregnancy-Related Deaths (Pregnancy through 42 Days Postpartum, Pregnancy-Related Only) (n)	Maternal Mortality Ratio (Per 100,000 Live Births)	Pregnancy-Associated Deaths (Pregnancy through 365 Days Postpartum, Any Cause) (n)	Pregnancy-Associated Mortality Ratio (Per 100,000 Live Births)
Total	39,720	15	37.8	32	80.6
Age (yrs)					
18–34	29,338	9	30.7	22	75.0
35–45	10,382	6	57.8	10	96.3
Unknown	0				
Ethnicity					
Non-Hispanic	34,046	15	44.1	29	85.2
Hispanic	4795	0	-	3	62.6
Unknown	879				
Race					
White	25,480	11	43.2	20	78.5
Non-White *	10,332	4	38.7	12	116.1
Unknown	3070				
Rurality					
Urban	28,887	14	48.5	29	100.4
Rural	10,613	1	9.2	3	28.3
Unknown	220				

Abbreviations: VA, US Department of Veterans Health Affairs; n, number; MMR, maternal mortality ratio; PAMR, pregnancy-associated mortality ratio; yrs, years. * Non-White includes Black/African American (9), American Indian/Alaska Native (2), Asian, and Native Hawaiian/Pacific Islander (1).

## Data Availability

Data are contained within the article and Appendix A.

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
