# Peer review of "Improving the Maternity Care Safety Net: Establishing Maternal Mortality Surveillance for Non-Obstetric Providers and Institutions"

_ijerph, 2023, doi:10.3390/ijerph21010037_

Round 1

Reviewer 1 Report

Comments and Suggestions for Authors

First, thank you for inviting me to review the paper "Improving the Maternity Care Safety Net: Establishing Maternal Mortality Surveillance for Non-Obstetric Providers and Institutions."

  The manuscript provides an intriguing analysis of a phenomenon, such as maternal mortality during pregnancy, in the first postpartum days, and within the first year of the infant's life, in a group defined as at risk by the Authors: the Veterans.   I am generally highly critical in order to be constructive and improve the quality of the paper, but in this case, I must acknowledge that the topic has been handled very competently and maturely, and I have almost nothing to say about the methodology.   Nevertheless, I have some suggestions for the following sections:   Introduction: Consider emphasising the importance of veteran experiences in the context of the national crisis. I suggest briefly discussing why Black or African American individuals and those living in rural regions are at heightened risk and the implications for healthcare. Furthermore, the section on Maternity Care Coordinators is crucial. Please explain how this programme assists veterans receiving care from dual healthcare systems and how it enhances the committee's goals. In addressing the highlighted challenges and enhancing the quality of treatment for pregnant veterans, stress the significance of your study aims.   Materials and Methods: Outline the research's inclusion and exclusion criteria, presenting them as bullet points. Describe the data sources and the validation procedure briefly. I would suggest giving further details about how the multidisciplinary maternal mortality review group worked. To guarantee transparency, please include a brief note on the Institutional Review Board (IRB) approval.   The paper's alarm signal is evident, among other things: mortality in this cohort is higher than in the general US population, as they explicitly mentioned in the "Discussion" section. However, it is obvious that there is a need to focus more on psychological support considering the cases studied's compromised mental well-being.

Author Response

  Response to Reviewer:

The manuscript provides an intriguing analysis of a phenomenon, such as maternal mortality during pregnancy, in the first postpartum days, and within the first year of the infant's life, in a group defined as at risk by the Authors: the Veterans.   I am generally highly critical in order to be constructive and improve the quality of the paper, but in this case, I must acknowledge that the topic has been handled very competently and maturely, and I have almost nothing to say about the methodology. 

Thank you!

Introduction: Consider emphasizing the importance of veteran experiences in the context of the national crisis. I suggest briefly discussing why Black or African American individuals and those living in rural regions are at heightened risk and the implications for healthcare. 

This has been added at the beginning of the introduction. (line 64 – 71)

Furthermore, the section on Maternity Care Coordinators is crucial. Please explain how this programme assists veterans receiving care from dual healthcare systems and how it enhances the committee's goals. 

This has been expanded to describe more about the important role of Maternity Care Coordinators. (line 119 – 124)

In addressing the highlighted challenges and enhancing the quality of treatment for pregnant veterans, stress the significance of your study aims.   

Materials and Methods: 

Outline the research's inclusion and exclusion criteria, presenting them as bullet points.

This has been added. (line 137 – 150)

Describe the data sources and the validation procedure briefly.

This has been further developed (line 151-158), (line 191-193)

I would suggest giving further details about how the multidisciplinary maternal mortality review group worked. 

This section has been substantially expanded. (line 212-221)

To guarantee transparency, please include a brief note on the Institutional Review Board (IRB) approval.  

As a quality improvement project at the VA, IRB review was not required. This has been explained more fully in the text. (line 133-135)

 The paper's alarm signal is evident, among other things: mortality in this cohort is higher than in the general US population, as they explicitly mentioned in the "Discussion" section. However, it is obvious that there is a need to focus more on psychological support considering the cases studied compromised mental well-being.

The role of mental health has been further emphasized in the conclusion section. (line 395-397)

Reviewer 2 Report

Comments and Suggestions for Authors

Dear authors, congratulations on the subject of study of the manuscript, as it is of importance to study of the cases of maternal mortality in veterans, as well as the factors that can lead to it

After reviewing the manuscript, I submit the following comments.

Best regards,

In sección abstract

The keyword “pregnancy-associated mortality”, “maternal outcomes”, “maternal mortality surveillance” are not a MeSH Term.

For this reason, it is recommended for greater publicity and ease of locating the article, within the databases, that it be replaced by other similar MeSH terms, where it is suggested that the keyword “maternal mortality surveillance” be replaced by “Epidemiologic Surveille”, as “maternal mortality” is already included in another keyword.

In the Introduction section

The information written in this section is very good, such as the mental health problems of veterans, the results of surveys carried out on The population of reproductive-aged veterans, etc., including maternal care with external health providers, breaking down health care in different health services

However, I detect a lack of mention of other articles that deal with segmented health care in other health systems, whether in the USA or other countries, for later comparison; Therefore, I suggest that you include at least one article that addresses it.

In the Materials and Methods section

First of all, at the beginning of this section it does not mention the type of study carried out, which it does mention in the abstract section. When reading the manuscript, I have observed that it is a retrospective study where the medical and social histories of deceased female veterans are analyzed. It should be mentioned at the beginning of the section.

I have had several doubts regarding the study, since it is not mentioned in this section.

It does not mention whether the target population is all the records of female veterans with an obstetric history or is only limited to a part of them, although it mentions the inclusion and exclusion criteria.

In the mortality review, what mortality diagnostic codes were used. It is recommended that you include them in an Annex, to better understand the study.

Another unresolved question is how they determined whether the veterans lived in a rural or very rural town. What criteria were used in said classification (economic factor, population density, geographic dispersion, etc.). The criteria used must also be included in the aforementioned annex.

They have commented that they reviewed maternal mortality by a multidisciplinary review committee. What professional categories were included in said committee? Was it civil, military or mixed? These doubts should be clarified in the manuscript.

Author Response

Response to Reviewe

In sección abstract

The keyword “pregnancy-associated mortality”, “maternal outcomes”, “maternal mortality surveillance” are not a MeSH Term.

For this reason, it is recommended for greater publicity and ease of locating the article, within the databases, that it be replaced by other similar MeSH terms, where it is suggested that the keyword “maternal mortality surveillance” be replaced by “Epidemiologic Surveille”, as “maternal mortality” is already included in another keyword. 

Thank you. This mesh term has been added (and maternal mortality surveillance has been deleted.) Two other mesh terms have also been added: pregnancy outcomes and Pregnancy high risk. (line 60-61)

In the Introduction section

The information written in this section is very good, such as the mental health problems of veterans, the results of surveys carried out on The population of reproductive-aged veterans, etc., including maternal care with external health providers, breaking down health care in different health services

Thank you

However, I detect a lack of mention of other articles that deal with segmented health care in other health systems, whether in the USA or other countries, for later comparison; Therefore, I suggest that you include at least one article that addresses it.

In the Materials and Methods section

First of all, at the beginning of this section it does not mention the type of study carried out, which it does mention in the abstract section. When reading the manuscript, I have observed that it is a retrospective study where the medical and social histories of deceased female veterans are analyzed. It should be mentioned at the beginning of the section. I have had several doubts regarding the study, since it is not mentioned in this section.

 This has been added. (line 125 – 127)

It does not mention whether the target population is all the records of female veterans with an obstetric history or is only limited to a part of them, although it mentions the inclusion and exclusion criteria.

This has been clarified to state that all pregnancies within the entire veteran population were analyzed. (line 139)

In the mortality review, what mortality diagnostic codes were used. It is recommended that you include them in an Annex, to better understand the study.

These are uploaded with this revised submission.

Another unresolved question is how they determined whether the veterans lived in a rural or very rural town. What criteria were used in said classification (economic factor, population density, geographic dispersion, etc.). The criteria used must also be included in the aforementioned annex.

This has been clarified (line 179 – 181)

They have commented that they reviewed maternal mortality by a multidisciplinary review committee. What professional categories were included in said committee? Was it civil, military or mixed? These doubts should be clarified in the manuscript.

This has been expanded and clarified. (line 206 – 209)

Reviewer 3 Report

Comments and Suggestions for Authors

Overall, this is a good paper. I enjoyed reading it.  But I recommend you consider expanding the introduction a little bit by including a few examples of what you meant by "upstream and downstream factors" that could impact pregnancy outcomes. Refer to page 2, line 51. 

Comments on the Quality of English Language

Overall, this is a good paper. I enjoyed reading it.  But I recommend you consider expanding the introduction a little bit by including a few examples of what you meant by "upstream and downstream factors" that could impact pregnancy outcomes. Refer to page 2, line 51. 

Author Response

Response to Reviewe

Overall, this is a good paper. I enjoyed reading it.  But I recommend you consider expanding the introduction a little bit by including a few examples of what you meant by "upstream and downstream factors" that could impact pregnancy outcomes. Refer to page 2, line 51. 

This language has been expanded and is more descriptive. (line 61-62) 

Round 2

Reviewer 2 Report

Comments and Suggestions for Authors

Dear authors, congratulations on the improvement of the manuscript.

However, after reviewing the manuscript, I submit the only comment, already commented in the previous review.

Best regards,

In the Introduction section

The information written in this section is very good, such as the mental health problems of veterans, the results of surveys carried out on The population of reproductive-aged veterans, etc., including maternal care with external health providers, breaking down health care in different health services

Author Response

Thank you very much for your feedback, time, and support reviewing our revision of this article and for bringing up the issue of segmented care and its relationship to maternal outcomes. We note this threat to high quality care and reference this risk to our specific population in a couple of ways:

  1. Risk of disruptions in care coordination between VA and community care. Please see reference:
  1. # 15 Kroll-Desrosiers AR, Crawford SL, Moore Simas TA, Clark MA, Mattocks KM. Bridging the Gap for Perinatal Veterans: Care by Mental Health Providers at the Veterans Health Administration. Women's Health Issues. 2019/05/01/ 2019;29(3):274-282. doi:10.1016/j.whi.2019.02.005
  1. Risk of community providers not being well trained in caring for the unique needs of veterans. Please see references
  1. #16 Tanielian T, Farmer CM, Burns RM, Duffy EL, Setodji CM. Ready or Not? Assessing the Capacity of New York State Health Care Providers to Meet the Needs of Veterans. RAND Corporation; 2018.
  2. # 17. Vest BM, Kulak J, Hall VM, Homish GG. Addressing Patients' Veteran Status: Primary Care Providers' Knowledge, Comfort, and Educational Needs. Fam Med. Jun 2018;50(6):455-459. doi:10.22454/FamMed.2018.795504

Because the population of veterans receiving care at the VA with referrals to community providers is a unique model within the US healthcare system, and because the population of veterans carries very specific risk factors related to their military service, we have used these targeted references that relate directly to our population. We  have not found other references from a more global perspective that reflect as directly on this issue.  If you have a particular reference in mind, we are happy to entertain including it in our manuscript. We are happy to discuss further and thank you for your time and attention to detail with our manuscript.